# Neglected Zoonotic Diseases: Advances in the Development of Cell-Penetrating and Antimicrobial Peptides against Leishmaniosis and Chagas Disease

**DOI:** 10.3390/pathogens12070939

**Published:** 2023-07-15

**Authors:** Sara M. Robledo, Silvia Pérez-Silanes, Celia Fernández-Rubio, Ana Poveda, Lianet Monzote, Víctor M. González, Paloma Alonso-Collado, Javier Carrión

**Affiliations:** 1Programa de Estudio y Control de Enfermedades Tropicales PECET, Facultad de Medicina, Universidad de Antioquia, Medellín 050010, Colombia; robledo@udea.edu.co; 2Department of Pharmaceutical Technology and Chemistry, ISTUN Instituto de Salud Tropical, IdiSNA, Universidad de Navarra, 31008 Pamplona, Spain; sperez@unav.es; 3Department of Microbiology and Parasitology, ISTUN Instituto de Salud Tropical, IdiSNA, Universidad de Navarra, 31008 Pamplona, Spain; cfdezrubio@unav.es; 4DNA Replication and Genome Instability Unit, Grupo de Investigación en Biodiversidad, Zoonosis y Salud Pública (GIBCIZ), Instituto de Investigación en Zoonosis-CIZ, Facultad de Ciencias Químicas, Universidad Central del Ecuador, Quito 170521, Ecuador; apoveda@uce.edu.ec; 5Department of Parasitology, Institute of Tropical Medicine “Pedro Kourí”, Apartado Postal No. 601, Marianao 13, La Habana 10400, Cuba; monzote@ipk.sld.cu; 6Grupo de Aptámeros, Departamento de Bioquímica-Investigación, IRYCIS-Hospital Universitario Ramón y Cajal, Carretera de Colmenar Viejo Km. 9.100, 28034 Madrid, Spain; victor.m.gonzalez@hrc.es; 7Department of Animal Health, Faculty of Veterinary Science, Complutense University of Madrid, 28040 Madrid, Spain; palalo02@ucm.es

**Keywords:** antimicrobial peptides, cell-penetrating peptides, intracellular pathogen, *Trypanosoma*, *Leishmania*

## Abstract

In 2020, the WHO established the road map for neglected tropical diseases 2021–2030, which aims to control and eradicate 20 diseases, including leishmaniosis and Chagas disease. In addition, since 2015, the WHO has been developing a Global Action Plan on Antimicrobial Resistance. In this context, the achievement of innovative strategies as an alternative to replace conventional therapies is a first-order socio-sanitary priority, especially regarding endemic zoonoses in poor regions, such as those caused by *Trypanosoma cruzi* and *Leishmania* spp. infections. In this scenario, it is worth highlighting a group of natural peptide molecules (AMPs and CPPs) that are promising strategies for improving therapeutic efficacy against these neglected zoonoses, as they avoid the development of toxicity and resistance of conventional treatments. This review presents the novelties of these peptide molecules and their ability to cross a whole system of cell membranes as well as stimulate host immune defenses or even serve as vectors of molecules. The efforts of the biotechnological sector will make it possible to overcome the limitations of antimicrobial peptides through encapsulation and functionalization methods to obtain approval for these treatments to be used in clinical programs for the eradication of leishmaniosis and Chagas disease.

## 1. Introduction

Protozoan parasites of the genera *Leishmania* and *Trypanosoma* comprise a diverse group of unicellular eukaryotic species that are distributed worldwide and are responsible for serious infections in animals and humans. Chagas disease is a zoonosis that is expanding from the Americas to countries around the world because of migratory patterns. Although the disease can be resolved in the acute phase, the development of the chronic phase can lead to complications from cardiomyopathy, arrhythmias, and megaviscera in 40% of cases. As expected, this fact has a significant effect on morbidity and mortality in the local population [1]. Cutaneous and visceral leishmaniosis are the most common clinical forms of the disease. Most are zoonoses, except for infections caused by *Leishmania donovani* and *L. tropica*, although recent studies have indicated the existence of animal reservoirs. The visceral form is fatal if not properly treated, and the cutaneous form leads to social stigma due to the development of skin lesions. In the same way, other less common forms, such as mucocutaneous and diffuse cutaneous leishmaniosis, lead to disfigurements that require medical intervention in a timely and adequate manner. It is estimated that about 1 million new cases of leishmaniosis are reported in nearly 100 endemic countries per year [2]. These zoonotic agents promote intracellular infections accompanied by virulence strategies and evasion of host immune defenses, which hinder the accessibility of the corresponding therapies and promote the occurrence of adverse effects. Consequently, treatment strategies based on selective drugs directed toward infected target cells represent a suitable approach to reducing the limitations of conventional therapies [3,4,5].

The abuse of antibiotics in the conventional treatment of microbial infections (not only bacterial but also protozoan parasitic infections), favors cumulative toxicity and resistance to antimicrobials, which forces the development of new antibiotics or new strategies for the control of these infections. Currently, the multiresistance generated in microbial agents against various treatments is reducing the success rate of antibiotics and has increased morbidity and mortality rates worldwide. In 2015, a Global Action Plan on Antimicrobial Resistance was created to mitigate this situation [6]. In fact, the World Health Organization (WHO) considers resistance to antimicrobials one of the main public health threats facing humanity. The magnitude of the problem impairs the achievement of some Sustainable Development Goals (SDGs) such as an end to poverty (SDG 1) and health and well-being (SDG 3), affecting the survival of more than 10 million people by the year 2050 [7]. This demonstrates that the development of innovative strategies as an alternative to the substitution of antibiotics is a first-order socio-sanitary priority. 

Remarkably, a group of natural peptide molecules that can interact with cell membranes and participate in relevant biological processes has attracted the interest of researchers in recent years. These are promising strategies to increase their therapeutic efficacy, avoiding the development of toxicity or resistance in a controlled way [8]. The members of this group of peptides share similar physicochemical properties, but their mechanisms of action can vary, thus making it possible to differentiate them into two subgroups: antimicrobial peptides (AMPs) and cell-penetrating peptides (CPPs) [9,10]. This review presents the novelties of these peptide molecules, their potential to replace traditional antibiotics, and their activity and mechanisms of action against two of the most relevant neglected zoonoses: leishmaniosis and Chagas disease. Likewise, some biotechnological innovations using peptides with the potential for clinical application focused on neglected zoonotic diseases are described.

## 2. AMPs and CPPs and Neglected Zoonotic Diseases: An Overview

Currently, the WHO recognizes 20 highly prevalent diseases in tropical areas, which together are known as neglected tropical diseases (NTDs). Many of them constitute life-threatening zoonoses, which further aggravates their socio-sanitary repercussions. The most representative examples of neglected zoonoses are vector-borne diseases, such as the three main forms of leishmaniosis, which are caused by different species of the *Leishmania* genus, as well as Chagas disease, which is caused by the parasite *T. cruzi* [11]. These zoonoses have been neglected for a long time by governments and pharmaceutical companies because they do not foresee economic benefits in the establishment of control methods. The COVID-19 pandemic has further delayed the intervention agenda against these diseases. As a result, the control of NTDs is still inadequate and remains extremely difficult today [12]. Given this scenario, the WHO has presented a new 10-year plan to put an end to the suffering caused by neglected tropical diseases: a road map for neglected tropical diseases 2021–2030 [13]. 

The conventional treatment of these diseases has important limitations since many of the drugs are from the early- and mid-20th century, have limited efficacy in advanced stages of the disease, and are nonspecific and/or highly toxic [14]. Therefore, the goal of finding new starting points for developing new drugs to effectively treat and control these diseases is a priority [14,15]. To overcome these deficiencies, AMPs and CPPs could become promising alternatives for the pharmaceutical design and treatment of intracellular infectious diseases. Recent studies have reported interesting antimicrobial properties of these peptides against intracellular bacteria, viruses, and the protozoan genera *Leishmania*, *Trypanosoma*, and *Plasmodium*, the etiological agents of the main forms of leishmaniases, trypanosomiasis, and malaria, respectively [10,16]. 

AMPs are short peptide sequences of fewer than 50 amino acids that participate in the innate first line of defense against invading pathogens. AMPs have demonstrated antimicrobial activity against a variety of pathogens, such as viruses, bacteria, fungi, and protozoa [17]. Their mechanism of action usually consists of permeabilizing the membrane, although alternative mechanisms that affect the biochemical processes of the pathogen, such as destabilizing its membrane or interfering with its synthesis of proteins or nucleic acids, have also been observed. By following one of these mechanisms, AMPs lead to the death of the microbial cell. Currently, researchers have access to a large database dedicated to AMPs. The Antimicrobial Peptide Database provides information on 3569 AMP peptides from the six kingdoms of living beings as well as others of a synthetic nature [18]. Currently, several AMPs (polymyxins and daptomycin) are approved for antibacterial treatment by the Food and Drug Administration (FDA), and many other AMPs are in clinical development [8,19].

CPPs are relatively short peptides (fewer than 35 amino acids) whose main function is to offer the possibility of transporting a variety of versatile cargoes into cells. That is, they allow the distribution of various types of molecules: proteins, peptides, nucleic acids, liposomes, nanoparticles, and drugs. This group of natural or synthetically generated peptides has the ability to translocate across cell membranes; in many cases, they do so at low concentrations and without significantly damaging the cell membrane [10]. The entry mechanism can be through indirect endocytosis (energy-dependent) or direct entry (energy-independent), which can be associated with membrane toxic activity, or through both mechanisms simultaneously. The handicap of endocytosis is that the peptide–cargo complex is not retained inside the endosome but can be released to reach its destination. For this, it is essential that the CPP is equipped with molecules capable of lysing the endosome [20]. There is now a new version of an extensive database dedicated to CPPs [21] to analyze and develop CPP prediction methods. To date, no CPP or CPP-conjugate drugs have been approved by the FDA [22]. One aspect to optimize in relation to vaccines and drugs is adequate administration into the cell. In this context, CPPs have penetration capacity and are not toxic, and their peptide sequences can be modified to prevent their proteolysis and optimize their function [23].

## 3. How Do AMPs and CPPs Specifically Target Protozoan Parasites without Harming the Infected Mammalian Cell? Cell Entry Mechanisms

A critical obstacle that peptides must overcome to exert their action on intracellular parasites (such as *Leishmania* spp. or *T. cruzi*) in any of their stages is the complex system of membranes that must be crossed in order to enter the infected mammalian cell and the parasitophore vacuole and finally reach the parasite that resides inside [24]. Furthermore, the interior of the vacuole is a hostile environment with an acidic pH where the parasite’s proteases can inactivate drugs of a peptide nature. The success of therapies is directly related to a high toxicity toward the parasite and a low toxicity to the cells of the host organism; this is known as selective toxicity. In this section, we discuss some possible entry mechanisms that allow peptide molecules to pass through the host cell without damaging it before reaching the parasite to exert their antiparasitic activity. Two entry mechanisms into the cell can be differentiated for nutrients, pathogens, or particles, including AMPs and CPPs. These are direct fusion and endocytosis [25]. 

The first is based on the differences in the composition and charge of the cell envelopes (membranes and glycocalyx) of both parasites and host cells. This is a direct penetration mechanism that involves electrostatic interaction between peptides and membrane charges and does not require ATP. Direct penetration is a highly limited entry pathway for conventional drugs, preventing them from exerting their action on their target [26]. However, peptides can use this route of entry, accumulating locally in the parasite and reaching the necessary concentration to be effective [15]. The cell membranes of these pathogens have an outer hemilayer with a higher percentage of anionic phospholipids, providing a global negative charge to the surface. Conversely, the membrane of mammalian cells is a bilayer membrane mainly made up of zwitterionic phospholipids and thus has a net neutral charge. Rivas and colleagues described in detail that the composition of the trypanosomatid membrane is made up of negatively charged components, such as phosphatidylcholine, sphingolipids, and sterols [27]. However, the structure of the membrane and its variations associated with the cell stage of the parasite, as well as whether this influences the interaction with AMPs and/or CPPs, remains to be studied in detail.

The other entry mechanism is through endocytosis or vesicular trafficking. Trypanosomatids have complex glycocalyx made up of multiple lipids and proteins bound to phosphorylated sugars of different natures. Once more, the composition of the glycocalyx changes depending on the cell stage. Often, these molecules confer a negative charge to the surface; however, they seem to constitute a barrier to the entry of peptides rather than an entry gate. Thus, entrance by endocytosis is limited to the flagellar pocket, a specialized region devoid of glycocalyx that accounts for 5% of the cell surface [28,29,30]. Endocytosis is the cellular route developed by the cell for the trafficking of substances with up to five different variants. Notably, the usage of the different pathways depends on the combination of cell surface receptors and the size of the cargo to be internalized [25]. The detailed mechanism of AMPs and/or CPPs internalization is not fully clear, but some hints have been elucidated [15]. Parasite phospholipase (PLA_2_) is involved in regulating the vesicular traffic by modifying the membrane phospholipids and inducing membrane deformation to generate the vesicles [31]. Inhibiting the PLA_2_ activity with bromoenol lactone blocks the trafficking of proteins such as transferrin and albumin. On the other hand, the hemoglobin acquisition in *Leishmania* and *Trypanosoma* is very interesting [32]. These parasites are auxotrophic for heme groups, i.e., they are not able to produce heme groups, but their uptake is essential for parasite survival. The most probable source is the heme group from hemoglobin. *Leishmania* amastigotes express a high-affinity hemoglobin receptor (HbR) located in the flagellar pocket. Then, different Rab GTPase/SNARE proteins mediate hemoglobin’s internalization to the lysosome, where it is degraded, and the heme group is released and transported to its final intracellular destination [33,34]. Therefore, it has been proposed that hemoglobin trafficking pathways could be an unexploited approach for new therapeutic strategies. Indeed, 40 amino acids derived from the hemoglobin binding domain of HbR are able to block hemoglobin uptake, thus inhibiting parasite growth [35]. It is conceivable that like hemoglobin, AMPs and CPPs can be transported through these endocytic pathways regulated by the multiple Rab GTPase/SNARE family. AMP and CPP use the same entry mechanism pathway of some nutrients, such as hemoglobin, and other particles. Deciphering the components involved, the surface receptor, Rab GTPases, and SNAREs could constitute an interesting approach to identifying new strategies based on the use of AMPs and CPPs with enhanced target specificity.

## 4. AMPs and CPPs as Alternative Therapies to Conventional Drugs against Leishmaniosis and Chagas Disease

It is important to consider the parasitic intracellular lifestyle when proposing an adequate therapeutic approach. The trypanosomatids *Leishmania* spp. and *T. cruzi* are examples of obligate intracellular parasites. After their uptake by mammalian cells (especially macrophages in the case of *Leishmania* spp. and macrophages, fibroblasts, and epithelial cells in the case of *T. cruzi*), these parasitic protozoan flagellates lodge in their amastigote form within the acidified parasitophorous vacuole [36]. *Leishmania* amastigotes can multiply within host cells and cause their lysis. The released amastigotes are then responsible for systemic infection. In the case of *T. cruzi*, after multiplying, the intracellular amastigotes differentiate into trypomastigotes, lyse the host cell, and are released into the bloodstream to spread and invade other nucleated cells (cardiomyocytes or cells of the gastrointestinal tract) [24]. In treating these infections, the drug or bioactive compound should cross the macrophage cell and the parasitophorous vacuole membrane to reach the parasite. Otherwise, the local concentration inside the vacuole could be too low to be effective against the parasite. Moreover, the stability of the drug once inside the vacuole and under an acidic pH should be considered [37]. 

To date, there are no vaccines available to protect humans against Chagas disease or any of the main forms of leishmaniases (visceral, cutaneous, and mucocutaneous). In addition, the treatments of choice are conventional drugs, which include antimonials, miltefosine, and liposomal amphotericin B (AmBisome^®^) against some *Leishmania* species (due to the difference between the clinical forms of leishmaniosis, there is no universal treatment) and benznidazole (Bz) and nifurtimox against *T. cruzi* infection, are not completely effective at eradicating these parasitoses. Furthermore, conventional pharmacological therapies have been limited due to their toxicity and side effects [38,39,40,41]. At this point, AMPs and CPPs represent a very promising opportunity to overcome these limitations and develop new therapies [42]. Natural AMPs and CPPs have been isolated from amphibians, snakes, fish, insects, arachnids, and viruses [43]. AMPs represent an essential defense mechanism integrated into the immune system of vertebrates and invertebrates that can be active against various types of pathogens, including trypanosomatids [44]. These peptides have been classified into four groups on the basis of their structure (β-sheet, α-helical, extended, and loop) and their cationic and amphipathic characters [45]. Cationic peptides preferentially bind to the parasite membrane. This preference is presumably because mammalian cells contain zwitterionic phospholipids in their membranes, while conversely, the membranes of *Leishmania* parasites have an anionic nature. However, even though AMPs and CPPs have shown selective binding to the parasite membrane, some of these compounds exert less activity against amastigote forms [46,47,48,49]. This is likely due to the different surface charges in different *Leishmania* life stages. The promastigote form is strongly negatively charged, mainly due to the content of surface lipophosphoglycan (LPG, which covers more than 60% of the surface); instead, LPG is present at very low or non-detectable levels in the amastigote form [50]. The external covering of the amastigotes is primarily made up of glycoinositolphospholipids (GIPLs) [51], and the affinity of the peptides for the plasma membrane of amastigotes could be decreased [47]. Thus, it should be noted that these studies show that promastigotes’ sensitivity to AMPs is higher than that of amastigotes [52]. 

Another useful property of AMPs is their ability to stimulate the host immune response, which can help eliminate the infecting pathogen and promote the clearance of infected cells [15]. This broad-spectrum activity makes them particularly useful for treating infections caused by complex microorganisms, such as *Leishmania* or *T. cruzi*. Similarly, CPPs have some properties that make them a powerful alternative to conventional drugs, especially for intracellular pathogens [26,53]. Among these properties, we can highlight their biocompatibility, ease of synthesis, and controllable physical chemistry [54]. Interestingly, CPPs can cross cellular membranes without disrupting them, limiting their toxicity to the target cell [54]. The specificity and low associated toxicity suggest that CPPs constitute powerful tools to combat infectious diseases. Furthermore, there is currently interest in exploiting the use of CPPs for their ability to transport a wide variety of cargo molecules into cells through covalent or non-covalent binding. Therefore, another complementary scenario in which conventional drugs are conjugated with a reference CPP to produce a formulation with high absorption and parasiticide activity seems feasible; this strategy would be capable of effectively defeating this resistance and increasing the spectrum of susceptible trypanosomatids [10].

The synthetic pseudomonas-derived peptides KDEL and REDKL cause a severe disruption in the membrane and the loss of cytoplasmic components in *L. tarentolae* promastigotes [55,56]. Likewise, *L. tropica* promastigotes exposed to analogs of halictine-2 (a novel AMP from the venom of eusocial honeybees) showed pores on their surface with a significant collapse of the membrane [57], and jellein (an AMP derived from the royal jelly of honeybees) caused pore formation and changed the membrane potential in *L. major* promastigotes [58]. This effect was also produced by amphotericin B (AmB). This polyene antifungal is often used intravenously for systemic fungal infections and is currently the first-line medication for treating leishmaniosis (the visceral, cutaneous, and mucocutaneous forms) in some endemic regions, such as India [59]. AmB treatment was implemented in Bihar after large-scale resistance to pentavalent antimony therapy developed [60]. AmB has better selectivity for membranes containing ergosterol than for those containing cholesterol, leading to pore generation and membrane fragility. However, AmB induces nephrotoxicity in a huge percentage of patients, probably due to damage to the glomerular membrane [61]. The liposomal formulation of AmB (AmBisome^®^) reduces this toxicity, but it comes with a high cost of treatment. 

In addition to these mechanisms of action, AMPs have also been demonstrated to exert their leishmanicidal activity through other routes. Following membrane integrity disruption upon peptide treatment, mitochondria are among the most important intracellular targets. For instance, the full-length enterocin AS-48 induces mitochondrial damage to *Leishmania* spp. promastigotes [47]. *Pseudomonas* exotoxin-derived peptides also cause the depolarization of promastigote mitochondrial membranes [55]. Additionally, a lauric acid brevinin conjugate limited *L. major* promastigote proliferation by changing the mitochondrial potential [62], and the lethal effect of the recombinant plant-derived defensin Vu-Defr on *L. amazonensis* was the result of mitochondria membrane potential loss, among other mechanisms [63]. This mechanism of action involving mitochondrial electrochemical potential has also been detected after the exposure of parasites to reference drugs, such as miltefosine and paromomycin [64,65]. Miltefosine is a compound derived from phosphocholine first used as an anti-neoplastic drug that comprises the only oral drug for treating leishmaniosis. Paromomycin is an aminoglycoside antibiotic broadly used for treating Gram-negative bacterial infections and was introduced for the treatment of leishmaniosis in 2006. As a result of the depolarization of the mitochondrial membrane, a release of ROS to cytosolic space is produced, and consequently, changes in the ionic balance result in the induction of apoptosis. Just like miltefosine [66], AMPs are capable of inducing this form of programmed cell death in *Leishmania* parasites. For instance, a synthetic peptide carrying the core of the *Vigna unguiculata* defensin caused culture inhibition of *L. amazonensis* promastigotes by activating an apoptotic-like cell death pathway [67]. Since the loss of mitochondrial membrane potential is a key indicator for the initiation of programmed cell death, synthetic peptides, such as the *Pseudomonas* exotoxin-derived and modified halictine, as well as AMPs from the temporin family, have been demonstrated to induce apoptosis in parasites [55,56,57,68].

The development of combination chemotherapy against leishmaniosis may prevent drug resistance and shorten the duration of treatment, thus reducing the cost of therapy [60]. In recent years, the synergistic effect of leishmanicidal drugs with AMPs has been analyzed. For instance, the combination of the synthetic anti-lipopolysaccharide peptides (SALPs) 19-2.5 and 19-4LF with paromomycin and AmB enhanced their activity against *L. major* amastigotes in vitro [69]. Conversely, this synergistic effect was not observed when a modified halictine-2-derived peptide was employed in combination with these leishmanicidal treatments. Nevertheless, potassium antimony (III) tartrate (PAT) in combination with this peptide showed a synergistic antileishmanial effect against intramacrophage *L. tropica* amastigotes through an unknown mechanism [57]. As described, synergistic effects between leishmanicidal compounds and AMPs are not always detected. This is the case for curcumin, a natural compound derived from the dried ground rhizome of the perennial herb *Curcuma longa* Linn (commonly known as turmeric), which has anti-inflammatory, anticancer, antiprotozoal, antiviral, and antibacterial activity. Although curcumin shows synergistic effects when applied in combination with paromomycin and miltefosine [70,71], such desirable outcomes are not detected when curcumin is combined with the CM11 hybrid peptide against *L. major* promastigotes or amastigotes [72,73].

Today, there is global alarm concerning the development of multidrug resistance among microorganisms, and this has also been identified in *Leishmania* [74] and Chagas disease [75]. AMPs and CPPs have multiple mechanisms of action, making it difficult for pathogens to develop resistance [76]. Furthermore, resistance to AMPs and CPPs is complex since it involves important changes in the phospholipid composition of cell membranes, which can result in pleiotropic effects on transport and enzymatic systems, seriously threatening the survival of microorganisms [77].

## 5. Antiparasitic and Immunomodulatory Activity of AMPs and CPPs

The antiparasitic activity of AMPs and CPPs, as we will detail later, is carried out through different mechanisms, such as the rupture of the plasmatic membrane, the alteration of calcium homeostasis (excessive accumulation of intracellular Ca^2+^ interferes with metabolism, the disorganization of kinetoplast DNA. and the promotion of autophagy and cell death) [78]. CPPs cross the membrane and tend to accumulate directly within the cytoplasm to carry out their antiparasitic activity by interfering with enzymatic activity and nucleic acid synthesis. These peptides may be involved in antiparasitic activity as well as immunomodulatory functions, leading to proper regulation of the inflammatory response to reduce damage to different target organs and control infection [42,79]. 

In Table 1, in vitro studies on natural and synthetic AMPs and CPPs against *T. cruzi* and different *Leishmania* species are summarized. Many of them were obtained by synthetic routes, although they generally respond to primary structures in the same manner as naturally ocurring peptides. The sources of AMPs and CPPs with antimicrobial activity, can be divided into mammals (human host defense peptides account for a large proportion), plants, amphibians, microorganisms, and insects. However, it is notable that mainly organisms used as primary sources are not common hosts of *Leishmania* or *T. cruzi* parasites.

Once internalized, the peptides and/or the cargo they carry can exert their antiparasitic activity. Two main mechanisms of action have been proposed: transient destabilization of membranes and intracellular targets [79]. In the case of cationic peptides, it has been suggested that electrostatic interactions based on differences in the compositions of the envelopes of host cells and parasites constitute the determining factor allowing preferential binding to the latter, using the host cell membrane as a means of passage. Binding to the parasite membrane induces its destabilization, which can lead to lysis and a loss of membrane potential. By contrast, when the peptides are anionic, hydrophobic, and amphipathic, the mechanism is elusive. In other cases, the target may be intracellular, such as an enzyme, nucleic acid, or organelle (e.g., mitochondria) [80,81]. As most AMPs are cationic, one of the mechanisms of action of such compounds is the selective binding to the membrane of the parasites, causing membrane disruption and pore formation [82]. Those pores might be formed by the dimerization of the peptides within the membrane upon these electrostatic interactions [83]. As previously mentioned, the differences in the external membrane charge between *Leishmania* life stages are responsible for the dissimilarity in susceptibility to compounds between the promastigote and amastigote forms. The intracellular nature of amastigotes creates additional barriers to the leishmanicidal activity of peptides. In an attempt to increase our knowledge on this topic, delivery systems, combinations, and chemical conjugation strategies have been tested. Since lipopeptides produced by *Bacillus* species target the cytoplasmic membrane and form ion-conducting pores in the lipid membrane, these lipopeptides are endowed with cytotoxicity toward human cells, which limits their biomedical application. Their encapsulation in chitosan nanoparticles, which previously exhibited antileishmanial potential through direct intercalation into the parasitophorous vacuole, enhanced antileishmanial activity [84]. This improvement could be related to the progressive release of lipopeptides from the chitosan delivery system [85]. This system previously showed effectiveness against experimental cutaneous leishmaniosis using AmB as the incorporated drug [86]. Among the strategies used to increase the therapeutic index and reduce the toxic effects of currently available chemotherapy against leishmaniosis, nanocarriers stand out, showing potential as a site-specific drug delivery system [87]. 

### 5.1. Synthetic and Bioinformatic Tools

An important advantage of AMPs is their broad potential for synthetic modification [88]. The molecular characterization of AMPs makes it possible to generate synthetic derivatives by modifying the primary sequence in order to improve some properties related to target specificity, cytotoxicity, potency, stability, or the active site [76]. This flexibility could enable the development of AMPs with optimized therapeutic properties for the treatment of both diseases. Among the most promising scaffolds for drug development, AMPs have been explored as potent antimicrobials because of their versatility and almost unlimited sequence space. These molecules can be easily tuned to achieve broad-spectrum activity, specific activity, or cytotoxicity through changes in the amino acid residues that are part of their sequence [89]. These changes give rise to variations in the structural and physicochemical properties that are closely related to their antimicrobial activity [90]. Cytotoxic activity may be enhanced by changing the amino acids. The synthesis of peptides bearing the sequence responsible for the biological activity is also useful [63]. Additionally, among the different kinds of peptide modifications, fatty acid conjugation to potentiate antimicrobial activity has been a topic of interest. However, the results have not always been as expected [58]. Either way, AMPs are toxins produced by organisms, such as frogs or snakes, and can require complex and expensive purification processes to be used as therapeutic agents. The synthetic production of AMP can also be expensive and time-consuming, but this problem can often be overcome through solid-phase peptide synthesis [91].

In addition, bioinformatics is a useful tool for active peptide selection. The physicochemical properties, structure, and toxicity of peptides can be predicted using bioinformatic tools in order to detect antimicrobial regions and determine the charge, hydrophobicity, isoelectric point (pI), and peptide mass. Cationic peptides bio-inspired by natural toxins have been recognized as an efficient strategy for the treatment of different health problems. Selected peptide sequences were synthesized and tested against cancer cells, bacteria, and two *Leishmania* species [92]. Another choice is the generation of hybrid peptides. CM11, which consists of the N-terminal domain of cecropin A and the hydrophobic C-terminal domain of melittin, demonstrated the ability to kill *Leishmania major* promastigotes and amastigotes with no significant cytotoxicity to murine macrophages [49]. The production of recombinant peptides using cloning strategies has also been tested. The insect defensin rDef1.3 from *Triatoma pallidipennis*, a vector of *T. cruzi*, was produced by transformed *Escherichia coli* and purified using immobilized metal affinity chromatography. Then, its microbicidal activity was analyzed against trypanosomatid species, including two *Trypanosoma* species as well as *L. major* and *L. mexicana*. Recombinant defensin caused atypical morphology and proliferative activity reduction in *Leishmania* parasites [93].

### 5.2. AMPs and CPPs for Combatting Different Forms of Leishmaniosis

The attractive biological activities of AMPs are prompting active research in the therapeutic application of these agents to combat many infectious diseases [94]. The first reports of the effects of AMPs on *Leishmania* were published in 1998, with *Hyalophora* cecropin A [95], cecropin A (1–8)–melittin (1–18) (CAMEL) hybrid peptides [96], and components of the target microorganisms, such as macromolecules and organelles [43,97]. To date, several groups of AMPs and CPPs, such as cathelicidins, cecropins, defensins, dermaseptins, eumenitin, histatins, magainins, melittins, and temporins, among others, have been proven to have significant action against diverse *Leishmania* species [69,94]. In this sense, relevant reviews in recent decades [10,81] have highlighted this group as an exciting alternative for designing new pharmaceutical alternatives against leishmaniosis and have demonstrated the growing list of AMP and CPPs with antileishmanial activity [10]. Below, a compilation of previously published studies regarding the action of these molecules against *Leishmania* is discussed.

The problem faced by traditional drugs in crossing the protozoa membrane and accessing intracellular amastigotes is well known. AMPs are characterized by their high intracellular penetrability. Their antiprotozoal activity could be direct, altering membranes or focused on internal targets, including DNA, RNA and protein synthesis, the lysosomal bilayer, key enzymatic activities, and mitochondria [98]. In addition to their ability to permeabilize membranes, recent observations have shown that some peptides can also move to the cytoplasm of the microorganism cell and interact with intracellular targets, interfering with the cell wall, the synthesis of nucleic acids or proteins, and enzymatic activity [99]. The high penetrability of AMPs will undoubtedly contribute to a faster mode of action than traditional drugs. This aspect is especially important in Chagas disease, as it will prevent the disease from progressing to the chronic phase. In addition, as a result of their multiple mechanisms of action, AMPs could act synergistically with conventional drugs (such as Bz and nifurtimox) and other AMPs, leading to better treatment outcomes [100].

The phospholipase A2 (PPA2)-derived peptides are enzymes commonly present in the venom of organisms from all kingdoms; they have a natural origin and can hydrolyze phospholipids from cell membranes. Short peptides derived from PPA2 can cross the membrane, showing effective activity against *Leishmania* promastigotes and amastigotes [101,102]. Interestingly, these cationic peptides, rich in lysine, increase their affinity when lysine is substituted with arginine [103]. Another cationic peptide, tachyplesin, derived from the horseshoe crab (*Tachypleus tridentatus*) has potential against *Leishmania* spp. and *T. cruzi* [104,105,106]. Tachyplesin is a 17mer peptide with a net positive charge that interacts with the parasite membrane, seriously compromising its integrity. Bovine lactoferrin-derived peptide also has leishmanicidal activity, which resides in its ability to permeabilize the membranes of promastigotes and axenic amastigotes of *L. donovani* [107].

CPPs can be coupled to cargos and translocated into the cell with high efficiency. Cargos include drugs or biological molecules, such as DNA, antibodies, and proteins [108]. Since CPPs can be placed in the membrane of the target cell, they can penetrate and accumulate inside the intracellular compartments, reaching higher local concentrations and overcoming one of the limitations of common drugs [10,26]. This fact is very interesting, considering the limited concentrations that can be attained in the plasma with common soluble drugs, which are even more limited in intracellular compartments. A few examples have been reported in this regard. Tachyplesine peptides were able to transport the plasmid EGFP-N1 inside parasites, becoming fluorescent [106]. Another example is chiral cyclobutane, which contains cell-penetrating peptides. These peptides are highly selective for *Leishmania donovani* parasites compared with HeLa cells [30]. While *Leishmania* promastigotes were not sensitive to free doxorubicin, the toxicity dropped to <1 µM when conjugated with these peptides, revealing their potential as a vehicle. In addition, when conjugated with TAT (transactivator of transcription), doxorubicin also accumulated inside the parasite, although to a lesser extent than with cyclobutane-CPPs [30]. TAT is a positively charged peptide derived from the TAT protein of HIV-1. The TAT protein binds and activates RNA polymerase II during infection [109]. TAT is a CPP that can pull cargos across membranes in different systems [110]. In *Leishmania*, TAT facilitates the internalization and accumulation of the antiparasitic miltefosine and paromomycin drugs [111,112,113].

As previously described, an appealing characteristic of AMPs is their ability to exert microbicidal activity by more than one mechanism. The strong post-transcriptional control of gene expression in trypanosomatids makes *Leishmania* a highly sensitive target to foreign RNases. This is the case for the ECP (eosinophil cationic protein, a human antimicrobial protein), which comprises RNase activity. Recently, Abengózar et al. [46] observed that ECP-treated *L. donovani* promastigotes showed a degraded RNA pattern. This was in agreement with the relationship between the recruitment of eosinophils into *Leishmania* lesions and favorable evolution. Just as paromomycin induces protein synthesis inhibition in *L. donovani* promastigotes [64], it has been hypothesized that xenocoumacin acts similarly on *L. tropica* promastigotes [114]. Recently, the modulatory effect of AMPs on *L. major* amastigote gene expression was shown as an additional mechanism [69]. The induction of autophagic cell death in the protozoan pathogen *L. donovani* has been described as an AMP mode of action; for instance, indolicidin and two peptides derived from Seminalplasmin (SPK and 27RP) prompt programmed cell death pathways without affecting host cells [115]. 

In addition, nanodelivery strategies can enhance the activity of peptides. Thus, the frog skin-derived peptide dermaseptin, which has been shown to possess antileishmanial activity [116,117], was encapsulated into sub-micrometer Cry crystal proteins formed naturally by *Bacillus thuringiensis*, enhancing the target to macrophage lysosomes. The encapsulation of dermaseptin in Cry crystal proteins improved the leishmanicidal activity of dermaseptin in both in vitro and in vivo infection models [53]. 

Currently, researchers are focusing their attention on the immunomodulatory ability of AMPs and CPPs. For instance, synthetic peptides derived from *Limulus* anti-LPS factors (LALFs) 19-2.5 and 19-4LF reduced the parasite burden in vivo when topically administered to *L. major* BALB/c-infected mice by modulating the expression of host genes [69]. Although each peptide displayed its own pattern of cytokine modulatory activity, these peptides caused an increase in Th1 cytokine mRNA levels (IL-12p35, TNF-α, and iNOS) in both the skin lesion and the spleen. In addition, in skin lesions from *Leishmania*-infected mice treated with the peptide 19-4LF, a decrease in *IL-4* and *IL-6* gene levels was detected, in agreement with the reduction in the parasite burden in these samples [69]. Phylloseptin-1 (PSN-1), a peptide found in the skin secretion of the frog *Phyllomedusa azurea*, showed activity against *L. amazonensis* promastigotes [118] and amastigotes [119]. To understand the molecular changes associated with the leishmanicidal effect of PSN-1 against amastigotes, the levels of key cytokines (TGF-β, TNF-α, and IL-12) and the production of reactive species (H_2_O_2_ and NO) were assessed. The increase in TNF-α release caused by PSN-1 might have participated in the destruction of the amastigotes inside macrophages. The peptide was also observed to up- and down-modulate IL-12 p70 production in infected macrophages in a concentration-dependent manner. Probably, the immunomodulatory effect of the peptide favors the host instead of the parasite by decreasing the pathogenesis while the peptide kills the parasite [119]. Amphibians are one of the most abundant reservoirs of AMPs in nature, and their peptides have been explored against different species of *Leishmania* parasite. Dermaseptin, which was isolated from frog skin secretions of *Phyllomedusa* genera, has shown activity against *L. major* [104,120], *L. amazonensis* [121], *L. mexicana* [122], *L. panamensis* [104] and *L. infantum* [123], agents that cause cutaneous or visceral leishmaniosis and are endemic to the New and Old World. Other frog-derived peptides inhibit the growth of parasites, such as bombinins H2 and H4 [124] and temporins [125,126], at micromolar concentrations. Substantial efforts have been invested in cecropin A and melittin alone or in combination as hybrid molecules (CA-M). In this sense, shortened sequences [127], lipid N-terminals [128], and N-methylated Lys residues [129] showing relevant activity against *L. donovani* [96] and *L. pifanoi* [128] have been designed. In particular, our attention was drawn to interesting results displayed by plant thionins with IC_50_ values < 0.5 μM [130], making them among the most efficient antimicrobial peptides; they also showed activity against other human pathogens [131,132]. However, one of the probable limitations of some of the included studies is related to antileishmanial activity with respect to the stage/form of the parasite targeted. In general, a large number of studies limited their results to the axenic amastigote (i.e., macrophage-free) or promastigote forms, which are not relevant in human infection. In *Leishmania*, the intracellular amastigote form is consistently more resistant, and their growth in hostile intramacrophagic habitats and membrane surface compositions most likely account for the differences observed upon comparison with promastigotes and axenic amastigotes [107]. In addition, few studies have demonstrated the use of AMPs and CPPs in animal models of infection by *Leishmania* parasites. In this sense, the therapeutic potential of CA-M analogs against canine leishmaniosis has been observed on the basis of infection control through a decrease in the parasite burden and a reduction in disease symptoms [77]. The lauric acid-conjugated form of brevinin, a defensin isolated from skin secretions, was administered alone and in combination with the CpG motif to treat BABL/c mice previously inoculated in the hind footpad with *L. major* metacyclic promastigotes. It is known that CpG motif application is helpful for inducing a specific immune response in experimental models. Brevinin was subcutaneously administered, whereas the CpG motif was applied via the intraperitoneal route five times over 10 days. In this case, in the fifth week after the challenge, the groups that received lauric acid conjugated form of brevinin alone or in combination with the CpG showed showed a significant increase in footpad swelling compared with the group of mice treated with the reference drug Ambisome^®^, which was able to notably control footpad swelling [62]. However, the parasite load decreased in the popliteal lymph nodes adjacent to the infection site after peptide administration more significantly in groups treated with the brevinin–CpG combination. The production of cytokines in the spleens of mice treated with brevinin did not coincide with parasite replication control results since it was not, as expected, favorable for the Th1 response, which is traditionally accepted as necessary for cutaneous leishmaniosis healing to occur [62]. Along the same lines, the frog skin-derived peptide dermaseptin, which has leishmanicidal activity, was encapsulated into crystal proteins and tested against a mouse cutaneous model of infection caused by *L. amazonensis.* This encapsulation strategy enhanced the peptide efficacy in the in vitro and in vivo infection models. Parasites were inoculated in the hind footpad, and the formulated peptide was intralesionally administered every four days (a total of six times). Repeated injections inhibited lesion growth efficiently. Similarly, encapsulated dermaseptin decreased the parasite burden in the footpads, whereas free peptide administration was unable to reduce the number of amastigotes in the lesions [53].

**Table 1 pathogens-12-00939-t001:** AMP and CPPs with antiprotozoal activity against intracellular parasites *T. cruzi* and/or *Leishmania* spp.

Peptide Molecule	Source	Antiprotozoal Activity	Reference
Andropin	Synthetic	*L. panamensis* *L. major*	[104]
Anti-lipopolysaccharide factor	*Penaeus monodon* (marine crustacean)	*L. braziliensis*	[105]
BatxC	*Bothrops atrox* (snake)	*T. cruzi* (Y strain)	[133]
Bombinins H2 and H4	*Bombina variegata* (frog)	*L. donovani*	[124]
Cathelicidins (SMAP 29, PG-1)	Synthetic	*L. major* *L. amazonensis*	[134]
Cecropin A, D	*Drosophila* *Hyalaphora cecropia*	*L. aethiopica* *L. panamensis*	[69,95]
Cecropin A-melittin	Hybrid peptide	*L. donovani* *L. pifanoi*	[96,128]
Cecropin A, B, and P1	Synthetic	*L. panamensis*	[104]
		*L. major*	
		*T. cruzi* (Tulahuen strain)	[135]
Chyral cyclobutanes	Synthetic	*L. donovani*	[30]
Clavanin A	*Styela clava* (sea squirt)	*L. braziliensis*	[105]
CM11 (cecropin–melittin hybrid)	Synthetic	*L. major*	[49]
Cryptdin-1 and -4	*Macaca mulatta* (rhesus macaque)	*L. major* *L. amazonensis*	[134]
Ctn	*Crotalus durissus terrificus* (rattlesnake)	*T. cruzi* (Y strain)	[136]
Defensin	*Phlebotomus duboscqi* (sandfly)	*L. major* *L. amazonensis*	[69,137]
Defensin α1	Human	*T. cruzi* (Tulahuen strain)	[138,139]
Defensin (fragments D, P, B, Q, and E)	*Mytilus galloprovincialis* (mussel)	*L. major*	[140]
Dermaseptin	*Phyllomedusa sauvagii* (frog)	*L. mexicana* *L. panamensis* *L. major*	[104,122]
Dermaseptin 01	Synthetic	*L. infantum*	[123]
Dermaseptin 01, 02, 03, 04, 06, and 07	*Phyllomedusa hypochondrialis* (frog)	*L. amazonensis*	[121]
Dermaseptin S1 analogs	Synthetic	*L. major*	[120]
Dhvar4 (histatin 5 analog)	Synthetic	*L. donovani*	[141]
DS 01	*Phyllomedusa oreades* (frog)	*T. cruzi* (Y strain)	[142]
Enterocin AS-48	*Enterococcus faecalis*	*L. pifanoi*	[143]
Enterocin AS-48 homologs	Synthetic	*L. donovani*	[47]
Eumenitin	*Eumenes rubronotatus* (wasp venom)	*L. major*	[69]
Gomesin	*Acanthoscurria gomesiana* (tarantula)	*L. amazonensis*	[144]
Histatin 5 (L- and D-enantiomers)	Synthetic	*L. donovani* *L. pifanoi*	[141]
Hmc364-382	*Dpenaeus monodon* (shrimp)	*T. cruzi* (Y strain)	[145]
Indolicidin	Synthetic	*L. donovani*	[115]
Lactoferricin (17–30) Lactoferrampin (265–284) LFchimera	Bovine milk lactoferrin (domain N1)	*L. pifanoi* *L. donovani*	[107]
LTP2 α-1	*Hordeum vulgare* (barley)	*L. donovani*	[130]
M-PONTX-Dq3a[1-15]/[Lys]^3^-M-PONTX-Dq3a[1-15]	*Dinoponera quadriceps* (ant) synthetic modification	*T. cruzi* (Y strain)	[146,147]
Magainin Magainin analogs (MG-H1/H2) and F5W-magainin 2	*Xenopus laevis* (frog)	*L. braziliensis*	[69,105]
		*L. major*	
		*L. donovani*	
	Synthetic	*L. amazonensis*	[148]
Melittin	Bee venom *Apis mellifera*	*L. donovani* *L. infantum* *L. panamensis* L. major *T. cruzi* (CL Brener strain)	[69,104,149]
Mylitin A	*Mytilus edulis *(mussel)	*L. braziliensis*	[105]
NK2	Synthetic	*T. cruzi* (Tehuantepec strain)	[150]
Ovispirin	Synthetic	*L. major* *L. amazonensis*	[134]
p-Acl and analog p-AclR7	Synthetic	*L. amazonensis* *L. infantum*	[102]
Penaeidian-3	Whiteleg shrimp *Litopenaeus vannamei*	*L. braziliensis*	[105]
Rhesus	Synthetic	*L. major* *L. amazonensis*	[134]
Phylloseptin-1	Synthetic	*L. amazonensis*	[118]
Polybia-CP	*Polybia paulista* (wasp)	*T. cruzi* (Y strain)	[151]
PTH-1	*Solanum tuberosum* (potato)	*L. donovani*	[130]
Pr-1, 2, and 3	Synthetic	*L. panamensis* *L. major*	[104]
Pylloseptin 7	*Phyllomedusa nordestina* (frog)	*T. cruzi* (Y strain)	[152]
SALPs	Synthetic	*L. major*	[69]
Snakin-1	*Solanum tuberosum* (potato)	*L. donovani*	[130]
Seminalplasmin (SPK and 27RP)	Synthetic	*L. donovani*	[115]
StigA25	*Tityus stigmurus* (scorpion) Synthetic	*T. cruzi* (Y strain)	[153]
Tachyplesin	*Tachypleus tridentatus* (horseshoe crab)	*L. panamensis* *L. major* *L. braziliensis* *L. donovani* *T. cruzi* (Y strain)	[104,105,106,154]
TAT (48–57) peptide TAT (48–60) peptide TAT and polyarginine	TAT (transactivator of transcription) protein from HIV-1	*L. donovani* *L. infantum*	[30,111,112]
Temporins A and B	*Rana temporaria* (frog)	*L. donovani* *L. pifanoi*	[69,126]
Temporin-1Sa, 1Sb, and 1Sc	*Pelophylax saharica* (frog)	*L. infantum*	[125]
Temporizin-1	Synthetic	*T. cruzi* (Y strain)	[155]
Thionin α-1, α-2, and β type I	*Triticum aestivum* (wheat) *Hordeum vulgare* (barley)	*L. donovani*	[130]
[Arg]^11^-VmCT1	*Vaejovis mexicanus* (scorpion)	*T. cruzi* (Y strain)	[156]

### 5.3. AMPs and CPPs to Combat T. cruzi Infection

We carried out an exhaustive bibliographical review of studies that have described the anti-*T. cruzi* activity of AMPs and CPPs. It should be noted that in our search, as in the searches of recent reviews by other authors [82], we did not find any studies that involved the analysis of these peptides from plants with activity against *T. cruzi*. Additionally, despite the wide variety of studies on peptides with in vitro antiprotozoal potential, there is currently no evidence to support the existence of AMPs or CPPs with in vivo anti-*T. cruzi* activity. This fact demonstrates, once again, the difficulty of obtaining a definitive treatment for this neglected zoonosis. The development of effective treatments for *T. cruzi* infection remains a great challenge, and more research is needed to identify new therapies. 

The first report on the effects of an AMP against *T. cruzi* was described in 1988. In that year, Jaynes et al. [135] demonstrated that two analogs of cecropin B were 50% and 100% effective in killing *T. cruzi* trypomastigotes in vitro. These analogs varied only slightly from cecropin B in their amino acid sequence homologies, and the charge distribution, amphipathic, and hydrophobic properties of the natural molecule were conserved. Since then, many natural and synthetic AMPs have shown activity against *T. cruzi*.

*Bothrops atrox* is a snake of great medical importance in the Amazon. Its venom is specialized for killing prey in nature, but it is also a source of peptides with antiprotozoal potential. For example, batroxicidin (BatxC) is a CPP extracted from its venom with trypanocidal activity [133]. Compared with Bz, BatxC was able to significantly reduce the number of free amastigotes (*T. cruzi* strain Y) in one assay after 24 h of incubation. The authors proposed that BatxC could kill epimastigotes by ROS generation, pore formation, and cell membrane degradation, inducing necrosis. Another antiprotozoal peptide is crotalicidin (Ctn), which is obtained from the venom gland of the rattlesnake *Crotalus durissus terrificus*. The peptide has a high selectivity index (>200) and is active against all morphological forms of the *T. cruzi* Y strain. The studies carried out reveal that the mechanism of cell death induced by Ctn seems to be necrosis and late apoptosis. Furthermore, the peptide showed a higher selectivity for the parasite compared with Bz [136]. M-PONTX-Dq3a is an AMP isolated from the venom of the ant species *Dinoponera quadriceps*. Two fragments derived from this AMP (M-PONTX-Dq3a[1-15] and [Lys]^3^-M-PONTX-Dq3a[1-15]) have been demonstrated to possess trypanocidal activities similar to those of the parent peptide against all three forms of the *T. cruzi* Y strain but with lower toxicity, better bioavailability, and a lower cost of production [146,147]. Due to their reduced peptide lengths, both fragments reached the clinical application phase. The mechanism of action appears to be the induction of parasitic necrosis through plasma membrane disruption and mitochondrial DNA fragmentation. Stigmurin (StigA25) is an AMP obtained from the venom gland of the scorpion *Tityus stigmurus*. StigA25 is stable to variations in pH and temperature and has antiparasitic activity of close to 100% against the *T. cruzi* Y strain, although its mechanism of action is still unknown [153]. [Arg]^11^-VmCT1 is another antiprotozoal peptide isolated from the venom of the scorpion *Vaejovis mexicanus* with activity against the three developmental forms of *T. cruzi* through a necrotic mechanism of action [156]. Another marine CPP with anti-*T. cruzi* activity is tachyplesin-I. It is a host defense peptide from the horseshoe crab *Tachypleus tridentatus* with antileishmanial activity, as previously mentioned, and it even possesses anticancer properties [105]. Tachyplesin-I was more potent against trypomastigote forms of *T. cruzi* than epimastigote forms. Moreover, tachyplesin-I did not show any cytotoxic effect against Vero cells. Again, these differences might be explained by the distinct surface compositions of the parasite forms. According to Souto-Padrón [154], the epimastigote forms have the least negative surface charge of all the developmental stages of *T. cruzi*, whereas trypomastigotes have the most negative surface charge. In any case, the antiparasitic mechanism has not yet been described. A promising antichagasic hemocyanin fragment obtained from the *Penaeus monodon* shrimp is Hmc364-282 [145]. The peptide showed high selectivity against the epimastigote, trypomastigote, and amastigote forms of the *T. cruzi* Y strain, and was clearly more active and less cytotoxic than Bz. Necrosis appears to be the mechanism of action of the peptide. Polybia-CP is a wasp venom AMP that was reported to be a potent trypanocidal agent [151]. The peptide was able to inhibit the main developmental forms of *T. cruzi* with higher efficacy and less cytotoxicity than the standard Bz. The great efficacy of Polybia-CP against intracellular amastigotes confirmed its high penetrability into the parasite (the number of amastigotes decreased by 38% after 24 h of incubation). The mechanism of action by which Polybia-CP exerted its antichagasic activity was via an apoptosis-like process. The peptide did not damage the membrane of the parasite even at concentrations higher than its EC_50_. These characteristics make Polibya-CP an interesting scaffold for the development of novel anti-Chagas therapies. The AMP melittin is the main toxic component in *Apis mellifera* venom. In vitro assays demonstrated that melittin affected all *T. cruzi* (CL Brener clone) developmental forms at low concentrations (up to 1 μg/mL) with low toxicity in mammalian cells [149]. It has been suggested that the mechanism of action of melittin depends on the parasite form. Accordingly, the main mechanism of cell death in epimastigotes and amastigotes would be autophagy. Conversely, in the trypomastigote form, melittin could produce cell death via apoptosis. In any case, melittin does not appear to affect the plasma membrane of the trypanosome. The in vitro activity and the different mechanisms of action confirm the great potential of melittin for the development of new therapies against neglected diseases such as Chagas disease. Dermaseptin 01 (DS 01) is a 29-residue-long peptide isolated from the skin secretions of the frog *Phyllomedusa oreades*. Bioassays revealed that DS 01 is a potent anti-*T. cruzi* Y strain agent. The peptide induced the death of the parasites by membrane disruption and cell leakage [142]. Pylloseptin 7 is a natural AMP isolated from the secretions of the frog *Phyllomedusa nordestina* with 1296-fold higher antitrypanosomal activity than Bz [152]. The peptide targets the plasma membrane of *T. cruzi*, leading to cell death by permeabilization. Pylloseptin 7 is a promising scaffold for the design of new antichagasic drugs. Defensin-α1 is a biologically active human AMP with demonstrated in vitro trypanocidal effects. Reported assays have indicated that this human peptide kills *T. cruzi* Tulahuen strain tripmastigotes and amastigotes in a peptide concentration-dependent and saturable manner [138,139]. It seems that its mechanism of action consists of the formation of pores in the membrane and the induction of nuclear and mitochondrial DNA fragmentation, leading to the destruction of the parasite. NK-2 is a shortened synthetic peptide formed by the cationic core-region-comprising residues 39 to 65 of porcine NK-lysine. Although both natural NK-lysine and NK-2 are capable of killing trypomastigotes (Tehuantepec strain), NK-2 demonstrated greater safety for human cells [150]. In addition, NK-2 also inhibited the replication of intracellular amastigotes. Although studies have been carried out, the mechanism of action of NK remains unclear, although the peptide quickly permeabilizes the parasite’s plasma membrane in minutes. This indicates that the parasite’s plasma membrane is targeted by NK-2, making the peptide a potential trypanocidal drug. Temporizin-1 is a synthetic hybrid peptide containing the N-terminal region of temporin A (produced by *Rana temporaria*), the pore-forming region of gramicidin, and a C-terminus consisting of alternating leucine and lysine [155]. Temporizin-1 is an improved version of temporicin created by shortening the four residues related to the gramicidin ionic channel pore, the origin of the unique mode of action of the peptide. The trypanocidal effect of temporicin-1 was studied in *T. cruzi* Y strain epimastigotes and was found to be dose-dependent, improving the antitrypanosomal activity of temporizine and gramicidin with less cytotoxicity. Regarding the mode of action, temporicin-1 seems to produce alterations in the mitochondria and nuclear DNA, albeit, curiously, causing no alterations to the plasma membrane. Its toxicity is based on the differences between the compositions of mammalian cell membranes and trypanosome membranes. Temporizin-1 appears to form ion channels in mammalian cell membranes, generating low toxicity. However, its toxicity towards trypanosomes seems to be attributed to an intracellular effect rather than pore formation.

## 6. Biological Models to Evaluate the Activity of AMPs and CPPs

The development of safe, effective, and efficient therapies and vaccines depends on the selection of appropriate biological models and a proper understanding of the advantages and limitations of these models since a well-designed biological model provides a solid foundation for supporting good science and ensuring the most beneficial use of resources. Biological models include in vitro, ex vivo, and in vivo animal models that, in the case of infectious disease research, are intended to emulate the biological phenomenon of interest for a disease occurring in a human or animal. Biological models of infectious diseases, such as leishmaniosis and Chagas disease, allow in-depth research on the molecular mechanisms of the pathology, high-throughput studies on new drugs and genetic targets, and the visualization of the specific effects of new molecules on these microorganisms. Therefore, these models allow scientists to obtain a complete overview of these diseases and perform detailed and efficient studies of possible diagnostic methods and new therapeutic alternatives.

### 6.1. Cell Lines and Primary Cell Cultures for Cytotoxicity Assays

Over the years, several types of cell line and primary cell cultures have been used to determine the cytotoxic activity of new molecules in vitro. Table 2 summarizes the cell lines most widely used in cytotoxicity assays [157,158,159,160]. The specific protocol, medium, and nutrients to grow and culture each cell line can be found on the American Type Culture Collection (ATCC) web page (https://www.atcc.org/; accessed on 1 June 2023).

Nonetheless, considering the biology of the *Leishmania* spp. and *T. cruzi* parasites, the most recommended cell types to model the effects of therapies against both parasite species are those isolated from humans, such as U-937 tissue monocytes, because these cells meet the parasites at the site of the insect vector’s bite. Additionally, for *T. cruzi*, other recommended cells include the *Homo sapiens* aorta smooth muscle fibroblast-like cell line T/G HA-VSMC (ATCC^®^ CRL 199^TM^), the *Homo sapiens* normal esophagus epithelial cell line Het-1A (ATCC^®^ CRL-2692™), and the *Homo sapiens* normal colon epithelial cell line FHC (ATC^®^ CRL-1831™) because these are tissues for which the different strains of *T. cruzi* exhibit tropism.

In addition, for in vitro cytotoxicity tests, it is also advisable to carry out an initial screening for liver and kidney toxicity, for which the most recommended cells are the *Homo sapiens* liver epithelial-like cell line Hep G2 (ATCC^®^ HB-8065™) and the *Homo sapiens* kidney epithelial tissue 293T (ATCC^®^ CRL-3216™). Most of these cells are maintained in culture at 37 °C under a 5% CO_2_ and >95% humidity atmosphere in DMEM or RPMI media (except for FHC cells, which are cultured in a DMEM/F12 medium), supplemented with 5–10% heat-inactivated fetal bovine serum (FBS), 100 U/mL penicillin, 0.1 mg/mL streptomycin, or 50 μg/mL gentamycin. Because some cell lines may require other specific nutrients, following the American Type Culture Collection (ATCC) instructions (https://www.atcc.org/; accessed on 1 June 2023), for the handling of each cell line is recommended.

Primary culture cells include splenocytes, peritoneal macrophages, and bone marrow-derived macrophages or dendritic cells obtained from BALB/c mice and hamsters cultivated in RPMI-1640 media supplemented with 10% of FBS, 50 μg/mL gentamycin or 100 U/mL penicillin, and 0.1 mg/mL streptomycin. Cytotoxic activity is expressed as the concentration of AMP and CPP compounds capable of reducing the cell viability of cells by 50% after 48 h of treatment, expressed as the median cytotoxic concentration (CC_50_) or the median lethal concentration (LC_50_).

### 6.2. Antiparasitic Activity

In vitro antileishmanial activity is routinely evaluated using *Leishmania* promastigotes grown at 26 °C in Schneider’s insect medium supplemented with 10% (*v*/*v*) heat-inactivated fetal bovine serum plus an antibiotic mixture of penicillin and streptomycin. Like cells, many parasite species/strains are used to evaluate the antiparasitic activities of new molecules. In the case of *Leishmania* spp., a large number of described species are pathogenic for humans. For cutaneous leishmaniosis, the species that are most prevalent in specific regions are used, such as *L. braziliensis*, *L. panamensis*, *L. amazonensis*, and *L. mexicana* in Central and South America and *L. major* and *L. tropica* in Europe and Asia; for visceral leishmaniosis, *L. infantum* is used in America and *L. donovani* is used in Europe, Asia, and Africa. In some *Leishmania* species, there are strains expressing the green fluorescent protein (GFP) gene as well as the luciferase (LUC) gene.

Similarly, to evaluate the trypanocidal activity of a compound, in vitro assays are traditionally used with forms of *T. cruzi* epimastigotes cultured in an LIT hepatic tryptose infusion at 28 °C with 10% fetal bovine serum (FBS), 10 U/mL penicillin, and streptomycin. A 10 µg/mL concentration is usually incubated at pH 7.2 in the presence of the candidate compound to be evaluated, and it is collected during the exponential growth phase. For *T. cruzi*, the most frequently used strains are the Tulahuen strain Discrete Typing Unit (DTU) VI clones C2 and C4, which express the beta-galactosidase gene [157,159,161,162,163,164,165] or the luciferase gene Luc2-Tulahuen [166]; Y strain DTU II (wild-type [167,168] or expressing the firefly luciferase gene [161]); Sylvio X10 strain DTU I [158]; Dm28c strain DTU I [165]; RA strain DTU VI [169,170,171]; and VD strain DTU VI [172].

In both types of assays, the corresponding parasite forms are seeded and incubated with different concentrations of the candidate compound whose activity is being evaluated [173,174]. The EC_50_ is calculated on the basis of the mean percentage reduction in parasites compared with the untreated controls. The index of selectivity is calculated by dividing the LC_50_ or CC_50_ by the EC_50_ (IS = LC_50_ or CC_50_/EC_50_).

### 6.3. In Vivo Models

Experimental animal models are models in which the scientist induces a disease or condition in animals [175]. These models can be perfectly adapted to evaluate the efficacy of AMPs and CPPs in combatting neglected zoonoses. The One Health approach extends beyond conventional zoonotic disease control models to consider the interactions between human and animal health systems within their shared environments and the broader social and economic context. Thus, these study models are an essential part of the fight against neglected zoonoses [176], which are the subject of this review. In addition, it is necessary to carefully adjust to the international guidelines of the Care and Use of Laboratory Animals to achieve scientific progress, always considering the “Four R” (Reduction, Refinement, Replacement, and Responsibility) and other fundamental ethical considerations. Consequently, the use of experimental animals should be avoided in research areas in which alternative in vitro or in silico methods are available. The choice of the most suitable animal model must be undertaken on the basis of a process of careful analysis in order to combine animal welfare with scientific progress [177].

There are no standardized protocols for the evaluation of compounds in animal models, and researchers have described different protocols. The species most widely used as animal models of the main forms of leishmaniases and Chagas diseases are the Syrian golden hamster (*Mesocricetus auratus*) and BALB/c, C57BL/6, and albino Swiss mice [40,178,179,180,181,182]. The use of dogs (the main reservoirs of zoonotic visceral leishmaniasis VL caused by *L. infantum* in the Mediterranean area, Middle East, Asia, and Latin America) as experimental models has led to great advances in the development of vaccines and immunotherapy against VL [183]. Remarkably, dogs are a good model for chemotherapy studies not only for leishmaniosis but also for Chagas disease, and they may be indicated for preclinical trials of new treatments [184]. Dogs mimic human disease, as they are able to reproduce the clinical and immunological findings described in chagasic patients and can develop cardiomyopathy [182,185]. In contrast to rodent models of *T. cruzi* infection, all clinical stages of Chagas disease can be consistently reproduced in Syrian hamsters, including the chronic phase with cardiomyopathy [186].

## 7. Challenges to Overcome Regarding the Current Limitations of AMPs and CPPs

Below, we briefly discuss the disadvantages of AMPs and CPPs that still need to be overcome to consider in vivo applications and the development of clinical trials: (i) the degradation of AMPs and CPPs by proteases can occur in the bloodstream and the gastrointestinal system; (ii) the inactivation of AMPs and CPPs as a consequence of binding other proteins; (iii) low metabolic stability and oral absorption; (iv) AMP and CPP removal by rapid excretion through the kidneys and liver; (v) potential toxicity and immunogenicity; and (vi) high production costs [187].

Therefore, several drawbacks remain to be resolved for the use of AMPs and CPPS as therapeutic agents. Peptide molecules can be susceptible to degradation by proteases in the body, which can limit their effectiveness as therapeutic agents [188]. Considering their peptide nature, AMP and CPP molecules generally have low stability. Stability issues can be addressed by using specialized delivery methods, such as nanoparticles or liposomes, to enable them to reach their target sites in the body. However, this requires an increase in the complexity and cost of treatment, which is something to consider in the case of neglected diseases because it affects patients’ adherence [17,189]. Likewise, because of their low stability, peptide molecules are expected to have a short half-life in vivo, which means that they may need to be administered frequently and in high doses to maintain therapeutic levels in the body. This can increase the risk of toxicity, although it seems that these peptides are non-toxic to human cells and have minimal side effects. Furthermore, AMPs with clinical applications generally bind at low doses; they are part of the innate immune system and should thus be well tolerated by the body [190]. However, some AMPs may also be recognized as foreign by the immune system, leading to an immune response, which could reduce their effectiveness as therapeutic agents [191]. As mentioned above, a handicap for CPPs entering by endocytosis is their need to be equipped with molecules capable of lysing the endosome and reaching their target [20], thus overcoming the drawback of endosomal entrapment.

## 8. Emerging Biotechnological Tools: Future Directions

To face the challenges presented by the current limitations of AMPs and CPPs, different research teams have worked in different fields. Some have used chemical strategies based on glycosylation and lipidation to increase the efficiency and activity of conventional AMPs and CPPs [89]; others have reduced the size of drug carriers so that they have dimensions on the nanometric scale and, consequently, can be more easily internalized in cells and reach specific intracellular locations [192]. Among the different systems that allow the encapsulation of AMPs and CPPs are liposomes, dendrimers, solid-core nanoparticles, carbon, and DNA nanotubes [193]. A recent review analyzed different nanosystems as AMP delivery vehicles. The authors classified them on the basis of the nature of their skeleton: organic or inorganic [192].

These systems can be functionalized with different biomolecules (antibodies, cell-penetrating peptides, carbohydrates, and aptamers) that provide binding specificity to different cell types or locations. Of these, aptamers are attracting great interest due to their properties and characteristics [194]. Aptamers are single-stranded, folded nucleic acids (RNA or ssDNA) that are able to specifically recognize target molecules with high affinity. The term aptamer, derived from the Latin word “aptus”, which means to fit, was first introduced by the Nobel laureate J.W. Szostak and A.D. Ellington [195] when they described the in vitro selection of RNA molecules that bind specifically to a variety of organic dyes. Aptamers are selected through an in vitro process called SELEX (Systematic Evolution of Ligands by Exponential enrichment) developed by L. Gold and C. Tuerk [196]. The selected aptamer population may act like polyclonal antibodies. Subsequently, individual aptamers may mimic monoclonal antibodies. Compared with antibodies, aptamers possess various advantages: (i) they can be obtained against non-immunogenic proteins; (ii) they can be regenerated; (iii) they are stable under a wide range of environmental conditions; (iv) they can be chemically modified; (v) there is no need for cell culture or experimentation animals to produce them; (vi) they can be produced with high reproducibility; and (vii) they can be labeled with a great variety of fluorochromes.

The use of AMPs and CPPs in the treatment of intracellular pathogens has been limited by their instability in vivo and their low penetration capacity in mammalian cells. In this sense, aptamers have shown enormous potential to target other drugs, something that could be of great interest in the case of AMPs and CPPs, as it would favor their clinical application. Thus, Lee et al. demonstrated that HPA3PHis, an AMP loaded in a conjugate of gold nanoparticles and DNA aptamers, was effective against *Vibrio vulnificus* infection in mice. These bacteria may cause disease in those who eat contaminated seafood or have an open wound that is exposed to warm seawater containing the bacteria. Ingestion of *V. vulnificus* can cause vomiting, diarrhea, and abdominal pain. In this study, the authors showed the ability of a conjugate of gold nanoparticles and DNA aptamers loaded with AMP to achieve complete inhibition of colonization in different organs, leading to a 100% survival rate among treated mice compared with control mice, which died within 40 h of infection [197,198]. In several studies, CPPs and aptamers were used for the development of anticancer therapeutic systems. In a recent study, the authors demonstrated that ST21 (cell penetrating peptide-modified aptamer) nanoparticles could be an opportunity for the simultaneous administration of chemical and genetic drugs to combat hepatocellular carcinoma [199]. In summary, considerable efforts have been directed toward the development of CPPs in combination with aptamers for their potential application in the treatment of bacterial infections and cancer. However, their potential use in the treatment of parasitic diseases has not yet been shown. In this sense, the use of specific aptamers against *Leishmania* [200,201,202,203] or *Trypanosoma* proteins [204,205,206,207] could occur in conjunction with AMPs and CPPs to treat this neglected zoonosis, similar to what has been described in cancer models. However, this option is not yet a reality.

## 9. Conclusions

Considering the priorities established by the WHO in relation to the need to achieve affordable, safe, and efficient treatments for two of the most relevant neglected zoonoses, leishmaniosis and Chagas disease, AMPs and CPPs represent a potent alternative to conventional antibiotics. Several studies have demonstrated the activity of these natural peptides against intracellular trypanosomatids [81,82,208]. Among the distinguishing features with respect to conventional therapies, it is worth mentioning that AMPs and CPPs present little or no toxicity to mammalian cells and exert few anti-inflammatory effects [8,15]. The high biomedical potential of these peptides is becoming apparent. Several AMPs have been approved for antibacterial treatment (polymyxins and daptomycin) by the FDA, and many other AMPs are in clinical development. However, to date, no CPPs or CPP–drug conjugates have been approved by the FDA due to several drawbacks (toxicity, endosomal entrapment, immunogenicity, and in vivo stability issues) that, as we have mentioned, are currently being resolved [8,79].

Therefore, the optimization of AMPs and CPPs using emerging biotechnology and the achievement of conventional therapies combined with the appropriate properties of AMPs and CPPs will allow us to increase the potential of AMPs and CPPs by reducing toxicity and adverse effects and preventing the appearance of multiresistance. The effort being made by scientists around the world in the field of innovative therapies is commendable. Alternatives based on antimicrobial peptide molecules against neglected zoonoses will be a reality in a short period of time.

## Figures and Tables

**Table 2 pathogens-12-00939-t002:** The cell lines most frequently used to determine cytotoxicity.

Cell Type/Tissue, Product Code	Organism
Normal epithelial kidney cells VERO (ATCC^®^ CCL-81^TM^)	*Cercopithecus aethiops* (African green monkey)
Normal epithelial kidney LLC-MK2 (ATCC^®^ CCL-7^TM^)	*Maccaca mulata* (rhesus monkey)
Embryo fibroblast NIH/3T3 (ATCC^®^ CRL-1658^TM^)	*Mus musculus* (mouse)
Macrophages from reticulum cell sarcoma J774A.1 (ATTC TIB-67™)	*Mus musculus* (mouse)
Myoblast H9c2(2-1) (ATCC^®^ CRL 1446^TM^)	*Rattus norvegicus* (rat)
Macrophage DH82 cell line	*Canis familiaris* (dog)
Normal tissue lung fibroblast MRC-5 (ATCC^®^ CCL-171™)	*Homo sapiens* (rat)
Epithelial bone osteosarcoma U-2 OS (ATCC^®^ HTB-96™)	*Homo sapiens* (human)
*Tissue monocytes U-937* (ATCC^®^ CRL-1593.2TM)	*Homo sapiens* (human)

## Data Availability

Not applicable.

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
