# Peer review of "Neglected Zoonotic Diseases: Advances in the Development of Cell-Penetrating and Antimicrobial Peptides against Leishmaniosis and Chagas Disease"

_pathogens, 2023, doi:10.3390/pathogens12070939_

Round 1
Reviewer 1 Report
This paper describes a comprehensive review on the potential of antimicrobial and cell-penetrating peptides to inhibit the parasitic agents that cause Leishmaniasis and Chagas disease. The paper presents an overly optimistic view for the potential utilization of these peptides. A practical utilization of these peptides for Leishmaniasis and Chagas disease is still far from sight given the limited information on the mechanism of action and difficulty to direct the peptides to specific parasitic targets.
In my opinion, topic 7 “Challenges to overcome regarding the current limitations of AMPs and CPPs” should be expanded to give a better picture of the technical and mechanistic limitations to use peptides as target-specific tools for parasite inhibition.
Topic 8 “Emerging biotechnological tools: Future direction” should be reformulated. Isolation of a target-specific RNA and ssDNA aptamer is by itself a huge challenge. Therefore, the proposition that RNA and ssDNA aptamers can be developed for peptide delivery is out of context. Instead, the authors should cosider describing the potential of synthetic chemistry both for peptide synthesis, modification, functionalization, coupling and, so on for practical applications.
Author Response
Dear reviewer.
We have tried to improve the manuscript following your recommendations:
- In my opinion, topic 7 “Challenges to overcome regarding the current limitations of AMPs and CPPs” should be expanded to give a better picture of the technical and mechanistic limitations to use peptides as target-specific tools for parasite inhibition.
We agree with the reviewer's point of view that the use of antimicrobial peptides described was too optimistic. We have revised the text to get a more realistic message regarding certain limitations pending to overcome. Please note that you may confirm the result in the new revised version.
- Topic 8 “Emerging biotechnological tools: Future direction” should be reformulated. Isolation of a target-specific RNA and ssDNA aptamer is by itself a huge challenge. Therefore, the proposition that RNA and ssDNA aptamers can be developed for peptide delivery is out of context. Instead, the authors should cosider describing the potential of synthetic chemistry both for peptide synthesis, modification, functionalization, coupling and, so on for practical applications
Since much of the topic of aptamers was out of context, we have decided to largely eliminate it and combine this section with other methodological approaches in chemical modifications and nanotechnology. Please note that you may confirm the result in the new revised version.
Reviewer 2 Report
This review would be beneficial for a wide range of readers across the fields. The topic is worthy to be explored, organization and writing style of the manuscript are well-accepted. However, there are some concerning points to make this article more complete. In the title, the two infectious diseases are highlighted, however, in introduction section the background information/significance of these diseases in pathogenesis, progression, treatment are not provided or provided in unclear manner. This will be difficult for the readers out of the field to follow the context, I suggest that some of these background contents should be provided (along with citations) prior to the results/discussions regarding that literature to enhance understanding of the readers. Additional minor comments are also provided below.
Details examples.
Title: I suggest change to: “Neglected zoonotic diseases: Advances in the development of cell-penetrating and antimicrobial peptides against Leishmaniosis and Chagas disease”
Introduction: Authors should write more about both infection diseases (Leishmaniosis and Chagas disease).
Section 5.3.: Please rephrase the second paragraph, it is very confusing and unclear.
Reference section: It should be checked for consistent format regarding the journal guidelines. Inconsistent use of full/abbreviated journal names are frequently found. Please check, correct, and standardize.
Author Response
Dear reviewer.
We have tried to improve the manuscript following your recommendations:
- Title: I suggest change to: “Neglected zoonotic diseases: Advances in the development of cell-penetrating and antimicrobial peptides against Leishmaniosis and Chagas disease”
Thanks for the suggestion. We accept your proposal, the final title is "Advances in the development of cell-penetrating and antimicrobial peptides against Leishmaniosis and Chagas disease"
- Introduction: Authors should write more about both infection diseases (Leishmaniosis and Chagas disease).
We agree that it is necessary in the introduction section to detail a little more in relation to these diseases in terms of pathogenesis, progression, and treatment aspects. We have included in the Introduction section the following lines in this regard:
- Introduction
Protozoan parasites of the genera Leishmania and Trypanosoma comprise a diverse group of unicellular eukaryotic species that are distributed worldwide and are responsible for serious infections in animals and humans. Chagas disease is a zoonosis expanding from the American to countries around the world because of migratory patterns. Although patients can resolve the acute phase, the development of the chronic phase can lead to complications from cardiomyopathy, arrhythmias, and megaviscera in 40% of cases. As expected, this fact has a significant effect on morbidity and mortality in the local population [1]. The cutaneous and visceral leishmaniosis are the most common of the clinical forms of the disease. Most are zoonoses, except for infections caused by Leishmania donovani and Leishmania tropica, although recent studies indicate the existence of animal reservoirs. The visceral form is fatal if not properly treated, and the cutaneous form leads to social stigma due to the development of skin lesions. In the same way, other less common forms, such as the mucocutaneous leishmaniosis and the diffuse cutaneous leishmaniosis, lead to disfigurements that require medical intervention in a timely and adequate manner. It is estimated that about 1 million new cases of leishmaniosis per year are reported from nearly 100 endemic countries [2]….
- Section 5.3.: Please rephrase the second paragraph, it is very confusing and unclear.
Following the reviewer's suggestion, we have rewritten the second paragraph to improve the reader's understanding, avoiding possible confusion. Indeed, the text in this section followed a somewhat confusing and overwhelming format. We have reduced text and linked ideas to get a better understanding of the descriptions. Please note that you may confirm the result in the new revised version.
- Reference section: It should be checked for consistent format regarding the journal guidelines. Inconsistent use of full/abbreviated journal names are frequently found. Please check, correct, and standardize.
We have reviewed the reference section and we are also waiting for possible modifications suggested by the editor of the journal in accordance with the journal guidelines. Likewise, we have removed some repetitive and unnecessary references. Please note that you may confirm the result in the new revised version.
Reviewer 3 Report
In this review work, Robledo SM et al. provide an extensive and detailed literature revision on the use of cell-penetrating and antimicrobial peptides against the parasites that cause Leishmaniasis and Chagas disease. Given the limited treatments available for these two neglected tropical diseases, the use of different approaches in attempting to reduce or eliminate the parasitic burden is crucial for the control of these diseases in the population. The review work carried out in this manuscript is of a great quality, grouping a considerable part of the knowledge there is about the use and mechanisms of action of these peptides. Therefore, I have only a few minor recommendations.
- I would recommend dividing the information within each section in different paragraphs to help the reader to identify and process the information more easily.
- On page 5, line 248 "pseudomona-derived KDEL and REDKL" appears in bold. It is not clear if this is meant to be highlighted in the text or if it is a typographical error.
- I would like to point out the distribution of the bibliographic references. For example, on page 8 line 367, up to 6 bibliographic references have been used for the first sentence of the section. In contrast, the information between lines 368-374 contains only one reference.
- Section 6.1 on page 16 contains very detailed information about cell lines and the conditions for their maintenance. The information regarding the lines could perhaps be included in a table and the information regarding their maintenance could be summarize using more bibliographic references.
- Line 834 on page 18 refers to Figure 1, but I did not identify this figure in the text.
- I would recommend summarizing section 8 about future directions.
Author Response
Dear reviewer.
We have tried to improve the manuscript following your recommendations:
- I would recommend dividing the information within each section in different paragraphs to help the reader to identify and process the information more easily.
The reviewer is right in the suggestion of trying to clarify the message through the corresponding paragraphs to help the reader not to get lost and overwhelmed by such a long text. We have divided the information by paragraphs and tried to improve the way of transmitting the message. Please note that you may confirm the result in the new revised version.
- On page 5, line 248 "pseudomona-derived KDEL and REDKL" appears in bold. It is not clear if this is meant to be highlighted in the text or if it is a typographical error.
True, it was a typographical mistake. We have corrected it.
- I would like to point out the distribution of the bibliographic references. For example, on page 8 line 367, up to 6 bibliographic references have been used for the first sentence of the section. In contrast, the information between lines 368-374 contains only one reference.
Once again, the reviewer's suggestion has improved the result, we have distributed the references more equitably without losing the veracity of what each author has reported in the corresponding citations.
- Section 6.1 on page 16 contains very detailed information about cell lines and the conditions for their maintenance. The information regarding the lines could perhaps be included in a table and the information regarding their maintenance could be summarize using more bibliographic references.
Following the reviewer's recommendations to clarify the presentation of the cell lines mentioned in section 6.1, we have prepared a table where all this information is synthesized. Please note that you may confirm the result in the new revised version.
In relation to information on the maintenance of cell lines we have added these lines in that section: “Most of these cells are maintained in culture at 37°C and 5% CO2 and >95% humidity atmosphere in DMEM or RPMI medium (except for the FHC cells which are cultured in DMEM: F12 medium) supplemented with 5% - 10% heat inactivated fetal bovine serum (FBS), 100 U/ml penicillin, 0.1 mg/ml streptomycin or 50 μg/mL of gentamycin. Nonetheless the specific protocol, medium and nutrients to growth and culture each cell line it is found at the American Type Culture Collection (ATCC) web page (https://www.atcc.org/)”
- Line 834 on page 18 refers to Figure 1, but I did not identify this figure in the text.
True, it was a typographical mistake. We have corrected it.
- I would recommend summarizing section 8 about future directions.
Following the reviewer's suggestion, we have summarized section 8 and we have tried to give a more applied meaning. Please note that you may confirm the result in the new revised version.